# The impact of face masks on emotion recognition performance and perception of threat

Melina Grahlow[1,2,3]*, Claudia Ines Rupp[4], Birgit Derntl[1,3,5,6]

1 Department of Psychiatry and Psychotherapy, University of Tübingen, Tübingen, Germany, 2 Graduate Training Centre of Neuroscience, University of Tübingen, Tübingen, Germany, 3 Tübingen Center for Mental Health (TüCMH), Tübingen, Germany, 4 Department of Psychiatry, Psychotherapy and Psychosomatics, Medical University of Innsbruck, Innsbruck, Austria, 5 Tübingen Neuro Campus, University of Tübingen, Tübingen, Germany, 6 Lead Graduate School, University of Tübingen, Tübingen, Germany

* melina.grahlow@med.uni-tuebingen.de

**Data Availability Statement:** All relevant data including supporting information are available from the Open Science Framework. Please find the relevant DOI here: https://doi.org/10.17605/OSF.IO/JY2ES.

## Abstract

Facial emotion recognition is crucial for social interaction. However, in times of a global pandemic, where wearing a face mask covering mouth and nose is widely encouraged to prevent the spread of disease, successful emotion recognition may be challenging. In the current study, we investigated whether emotion recognition, assessed by a validated emotion recognition task, is impaired for faces wearing a mask compared to uncovered faces, in a sample of 790 participants between 18 and 89 years (condition *mask vs. original*). In two more samples of 395 and 388 participants between 18 and 70 years, we assessed emotion recognition performance for faces that are occluded by something other than a mask, i.e., a bubble as well as only showing the upper part of the faces (condition *half vs. bubble*). Additionally, perception of threat for faces with and without occlusion was assessed. We found impaired emotion recognition for faces wearing a mask compared to faces without mask, for all emotions tested (anger, fear, happiness, sadness, disgust, neutral). Further, we observed that perception of threat was altered for faces wearing a mask. Upon comparison of the different types of occlusion, we found that, for most emotions and especially for disgust, there seems to be an effect that can be ascribed to the face mask specifically, both for emotion recognition performance and perception of threat. Methodological constraints as well as the importance of wearing a mask despite temporarily compromised social interaction are discussed.

## Introduction

The congenital and cross-cultural ability to recognise facial emotional expressions is considered a prerequisite for successful social interaction [1]. In times of a global pandemic, people are widely encouraged to wear face masks covering mouth and nose in order to minimise risk of infection. However, this preventive measure might crucially affect social interaction: As wearing a face mask leaves only the upper areas of the face visible, namely eyes and forehead,

**Funding:** We acknowledge support by Open Access Publishing Fund of University of Tübingen.

**Competing interests:** The authors have declared that no competing interests exist.

this raises the question whether covering lower facial areas might interfere with successful emotion recognition.

Emotions have a regulative social function and constitute an important factor in appropriate interaction between individuals: Emotional expressions instigate informative processes that influence individuals' behaviour during social interaction [2]. For instance, the happy facial expression of your counterpart during a conversation will most likely encourage you to keep telling that story, whereas an angry expression might let you quickly change the subject. The success of our social interaction depends on the accuracy of emotion recognition, as different emotions convey different information by facial cues, such as the movement of certain muscles in the eye, nose and mouth regions [3]. Prototypical (surgical) face masks as worn during a pandemic to reduce the risk of infection, may reduce the likelihood of accurate emotion recognition. These face masks typically cover mouth and nose and leave only the upper areas of the face visible, which might undermine the success of our social interactions by reducing the number of available facial cues. The hindering effect of face masks on accurate emotion recognition may be especially relevant in interpersonal relations at the workplace where appropriate social interaction is expected. Furthermore, concerns on how face masks might (harmfully) impact child development have been voiced and it has been suggested that certain communication techniques might be important to ensure effective communication with children further on [4]. Effects of face masks are also relevant in the context of mental disorders: Individuals with emotional processing alterations such as e.g. individuals with autism spectrum disorder have been shown to employ perceptual strategies that are not optimal for face-processing, such as focussing mostly on the mouth region instead of directing their gaze to the eye region as well [5]. It becomes clear that not only specific groups of individuals such as those with a mental disorder but each and every one of us depends on successful social interaction, which is why it is important to investigate the possibly detrimental effect of face masks on emotion recognition performance across the general population.

To investigate emotion recognition in faces, participants are typically presented with photographs of emotional expressions [6] and are asked to judge the respective emotion. Most studies investigate the recognition of basic emotions, such as anger, fear, happiness, sadness and disgust [7]. Recognition accuracy has been found to generally differ between emotions: Happiness, the only positive of the basic emotions and hence probably the most distinct, has been found to be the emotion easiest to recognise [8–10]. Fear, on the other hand, seems to be recognised less accurately than other basic emotions [9, 11]. Further, there have been reports of systematic mistakes in emotion recognition with fear and surprise [9], as well as anger and disgust [12] being frequently confused. This may be due to the same facial action units being active during expression of several emotions, e.g. lowered eyebrows and narrowed eyes during expression of both anger and disgust [13, 14].

Several studies have stressed the importance of facial areas in emotion recognition. During free inspection of faces, observers tend to focus mostly on the eye region [15]. Studies using the 'Reading the Mind in the Eyes Test' [16] show that the eye region is informative of the mental state, i.e., the feelings and thoughts, of a person and may thus pose an important facial area for emotion recognition. However, previous findings are inconclusive concerning which facial areas might be the most important for correct recognition of specific emotions: While emotion recognition is most accurate if participants are presented with full face displays, it was found to be less accurate when only the lower region of the face can be seen and the least accurate if participants only see the upper region of the face [17]. Furthermore, the importance of lower and upper facial areas for emotion recognition seems to depend on the emotion itself [7, 18]. The lower region of the face has been found to be superior in recognition of happiness, while the upper region of the face seems to be important for recognition of fearful and sad

expressions [19–21]. For recognition of anger, results remain inconsistent with some studies stressing the importance of the eye region [22] while others find lower recognition accuracy for anger upon occlusion of the mouth [23]. For disgust, it is neither solely the lower nor the upper region of the face but more the nose and cheek region that seems to yield the most important information for recognition [24]. More recent research investigating the importance of facial action units applied eye tracking and confirmed that observers strongly rely on the mouth region when recognising happiness and disgust but focus more on the eyes upon recognition of anger, fear and sadness [25]. However, literature on facial action units and which areas are especially important for emotion-specific recognition is not conclusive: The general pattern of which facial area is important for recognition of which emotion is inconsistent and seems to at least partly depend on the measure [26].

One factor that might influence emotion recognition is the participant's age. Research investigating age differences report inconsistent emotion-specific effects: while emotion recognition seems to be less accurate for fear [27, 28], anger [27–29], and sadness [28–30] in older participants (between 65–80 years of age), improved recognition with older age (between 58–70 years) has been observed for happiness [30] and disgust [29].

Considering that emotion recognition seems to be impaired if faces are partly occluded, this raises the question whether face masks could also influence how these equivocal facial expressions are perceived. As facial expressions convey information about the emotional state of a person and therefore have a crucial communicative function [31], possible behavioural intentions might be misread when wearing a mask. Further, faces covered by masks may be perceived as more threatening, not least because (surgical) face masks are usually worn in the context of disease and might therefore be interpreted as a health threat. For instance, it has been found that a surgeon's mask seems to interfere with the communication to the patient [32]: This could render certain situations to be perceived as more threatening due to impaired verbal and non-verbal communication about topics related to disease. In turn, this might leave the mask as a symbol for a generally more threatening environment. Furthermore, it has recently been found that surgical face masks have a detrimental effect on face matching performance [33], which might further add to an increased perception of threat due to the impaired ability to recognise both unfamiliar and even familiar faces. Prior research has shown that face masks curtail perceptions of closeness [34] and that masked faces are perceived as less trustworthy than unmasked faces [35]. Further, angry and neutral faces covered by a sanitary mask or a scarf were evaluated as more negative compared to uncovered faces, whereas emotional evaluations did not differ between the two conditions [36]. The effect specific to a face mask on perception of threat thus remains to be investigated more closely.

While there is a large amount of studies investigating emotion recognition in faces that are not occluded or in faces that are partly masked, using for example the Bubbles technique [37], there are only very few studies looking at the impact of face masks specifically on emotion recognition accuracy. It has been suggested that reading of facial emotions is irritated by the presence of a mask [35, 38] and that emotion recognition accuracy may decline by almost 20% for masked faces [34], however, this was tested in relatively small samples and further studies including a control group assessing mask-specific effects are necessary. Another study investigated the effect of face masks and sunglasses (an occlusion individuals tend to have more experience with) on emotion recognition and found that, while recognition performance was reduced in both cases, accuracy was even lower in the mask condition [39]. Further research comparing the effect of sunglasses and face masks on emotion recognition performance in children found that children were still able to make accurate inferences about the emotional expressions shown, even when the faces were partly covered by sunglasses or a mask [40]. While a small effect of face coverings on emotion recognition accuracy was shown, the

sunglasses and mask conditions did not differ significantly from each other. This remains to be investigated more closely by inclusion of further mask-related control conditions, as sunglasses normally only cover the eye region; face masks, however, cover the lower half of the face, thus both occlusion types potentially hindering recognition of different facial emotional expressions.

The aim of the present study thus was to investigate whether emotion recognition is more difficult for faces wearing a face mask covering mouth and nose than for uncovered faces. We assessed emotion recognition by use of an adapted version of a validated emotion recognition task ('Vienna Emotion Recognition Task', VERT-K; [41]), where we digitally added surgical face masks to the original facial stimuli (condition *mask vs. original*). We expected impaired emotion recognition, i.e., lower recognition rates, for masked faces compared to unmasked faces. In a control condition consisting of two parallel versions, we further investigated whether emotion recognition performance for faces covered by a face mask differs from recognition performance for faces where only the upper half is presented and for faces where a bubble occludes the lower facial regions (condition *half vs. bubble*). Additionally, we were interested in whether wearing a face mask affects the perception of threat and therefore included an exploratory part where we assumed that faces with a mask would be perceived as more threatening than faces that are fully visible. For half faces and faces covered by a bubble, we also investigated whether the perception of threat is affected.

Using convenience sampling, we recruited large samples of individuals who completed the tasks online, achieving our overall aim of investigating the influence of a (surgical) face mask on emotion recognition performance and perception of threat.

## Materials and methods

### Sample

The current study comprised two experimental conditions. Condition *mask vs. original* (with stimuli 'mask' and 'original'; please see Fig 1) was conducted by 790 participants (636 women) between 18 and 89 years ($M$ = 30.85, $SD$ = 12.68). Condition *half vs. bubble* consisted of two parallel versions using the stimuli 'half' and 'bubble', see Fig 1. Here, 395 participants (289 women) between 18 and 70 years ($M$ = 29.16, $SD$ = 11.12) saw 'half' faces and 388 participants (298 women) between 18 and 65 years ($M$ = 28.94, $SD$ = 10.83) were presented with 'bubble' stimuli.

Subjects were recruited through the universities' mailing lists and through advertisements in groups in social media (e.g. Facebook). Experiments were conducted online and the participants' data was collected anonymously. The study was approved by the ethics committee of the Medical Faculty of the Eberhard Karls Universität Tübingen (252/2020BO2) and all research was performed in accordance with the relevant guidelines and regulations. Participants took part voluntarily and provided written informed consent before participation.

### Materials

The study was conducted via online questionnaires that were generated using SoSci Survey [42] and were made available to users via www.soscisurvey.de between April 14 and June 14, 2020 (condition *mask vs. original*) and between September 14 and October 14, 2020 (condition *half vs. bubble*).

To assess emotion recognition in faces with and without a face mask, an adapted version of an emotion discrimination task ('Vienna Emotion Recognition Tasks', VERT-K [41]) was applied. The VERT-K is a computer-based task and consists of 36 coloured photographs of facial expressions of Caucasians portraying five basic emotions (anger, fear, happiness, sadness

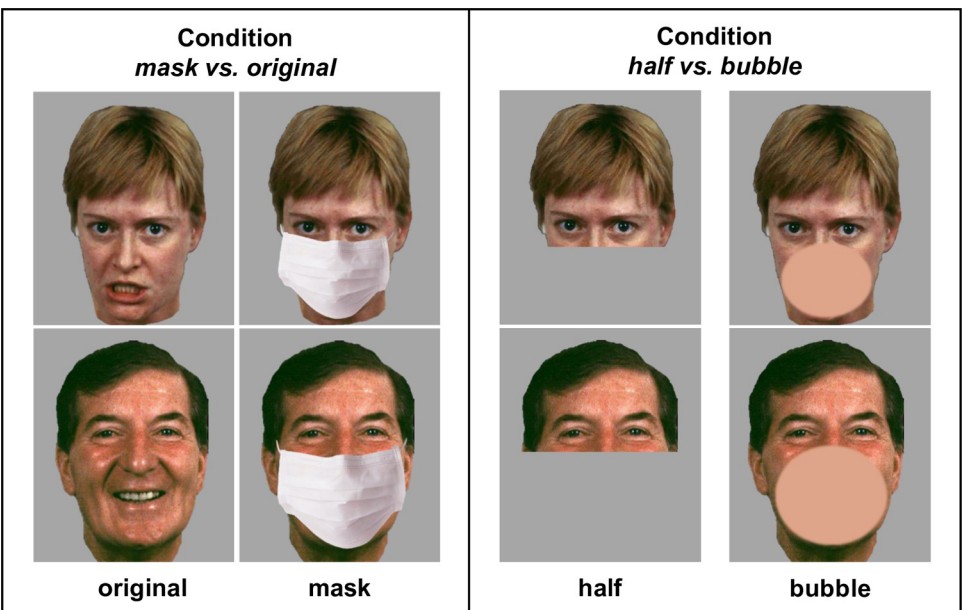

**Fig 1. Exemplary stimuli of the adapted Vienna Emotion Recognition Tasks (VERT-K).** A female poser showing an angry expression and a male poser showing a happy expression. Exemplary stimuli as used in the condition *mask vs. original* and control stimuli used in condition *half vs. bubble*. Images extracted and modified from [6].

and disgust) as well as neutral expressions. The photographs were originally taken from a standardised stimulus set (see [6] for development of stimuli) and have been validated for the German-speaking population [27]. The stimulus set consists of six evoked expressions per facial category, which are balanced for sex of poser (18 female and 18 male faces) with each poser appearing only once.

For condition *mask vs. original*, the VERT-K was extended by duplicating the stimulus set and by digitally adding a face mask covering mouth and nose to the duplicated faces, resulting in a total of 72 stimuli (36 without face mask and the same 36 with face mask). We chose an image of a typical white surgical face mask, fitted it to match the length and width of the respective face and superimposed the mask onto the original stimuli so that it covered the face from just below the eyes, i.e., onset of the nose, downwards. This was done using Adobe Photoshop [Adobe Inc., CA, USA]. The original black background of the stimuli was replaced with a grey background to reduce contrast to the white face masks and the size of the stimuli was standardised (14.50 cm x 12.70 cm). See Fig 1 for exemplary stimuli without ('original') and with face mask ('mask'). To ensure viability, the final 72 stimuli were split into two versions of 36 stimuli so that each poser was only seen once in each version (if poser A was wearing a face mask in version A, they were presented without a face mask in version B). Both versions were balanced for emotion, sex and age of poser, as well as face mask and stimuli were presented in a pseudo-randomised order making sure that no more than two masked faces were presented in a row and that the same emotion was not depicted more than three times in a row (see S1 Table 1 in S1 File). Participants were quasi-randomly assigned to one of the two versions.

For condition *half vs. bubble*, to assess emotion recognition in faces that are partly occluded by something other than a face mask, the 36 stimuli of the VERT-K were adapted in two different ways. For 'half' stimuli, the original stimuli were cut off just beneath the eyes so that only the poser's upper face was visible. For 'bubble' stimuli, a skin-toned bubble obscuring the

mouth and nose area of the poser was digitally added to the original images via Adobe Photoshop [Adobe Inc., CA, USA]. Again, the original black background was replaced with a grey background. See Fig 1 for exemplary stimuli used in this condition. Participants were quasi-randomly assigned to one of the study versions and were presented with one of the two adapted stimulus sets each consisting of 36 images in a pseudo-randomised order.

In both conditions, participants were instructed to recognise the respective emotion depicted and to select their answer in a forced-choice answering format where all categories were present (anger, fear, happiness, sadness, disgust and neutral). Each image was presented for an unlimited amount of time and the response options were presented at the same time as the image. There was no time limit for giving a response.

To assess whether faces wearing a mask are perceived as more threatening than faces without a mask, a Visual Analogue Scale (VAS) was used. Participants were asked to indicate how they perceived each face on a scale ranging from 0 (*not threatening at all*) to 100 (*extremely threatening*). For each item, the VAS was presented sequentially after forced-choice assessment of emotion recognition with the respective image still visible. There was no time limit for giving a response. At the end of the online questionnaires, participants provided demographic data such as age and sex.

## Statistical analyses

Statistical analyses were performed using SPSS 25 [IBM SPSS Statistics] with alpha set to .05. Despite minor deviations from normality (visual inspection and Kolmogorov-Smirnoff tests), we relied on parametric tests including analysis of variance (ANOVA) as these are adequate due to the reasonably large sample sizes [43].

Separately for each condition, the number of correct responses was calculated for each emotion, resulting in a mean score of emotion recognition accuracy (percent correct) for each participant per emotion for faces with and without mask (condition *mask vs. original*), for half faces and for faces with a bubble obscuring mouth and nose (condition *half vs. bubble*).

The current study aimed to investigate whether emotion recognition performance differs depending on the presence of a face mask (condition *mask vs. original*) or another type of occlusion of the lower facial areas (condition *half vs. original*). For this purpose, we relied on a *6 x 4* repeated-measures analysis of variance (ANOVA) with *emotion* (anger, fear, happiness, sadness, disgust and neutral) as within-subject factor and *type of occlusion* ('half', 'bubble', 'mask' and 'original') as between-subjects factor. For significant interactions, we performed separate emotion-specific ANOVAs.

To analyse whether participants perceive faces wearing a mask as more threatening than faces without a mask and to test for differences in perception of threat depending on the type of occlusion, individual ratings on the VAS were averaged, resulting in a mean score of perception of threat for each participant per emotion for each type of occlusion. These scores were analysed using a *6 x 4* ANOVA with the same factors as stated above. For significant interactions, separate ANOVAs were performed for each emotion.

For all ANOVAs, where sphericity was violated, Greenhouse-Geisser adjusted values are reported. Post-hoc tests with Bonferroni correction were calculated for significant results. We further report estimates of effect size for significant results using partial-eta-squared.

## Results

### Facial emotion recognition

Fig 2 depicts the descriptive statistics of the performance in the facial emotion recognition task for each type of occlusion. The ANOVA revealed a significant interaction between emotion

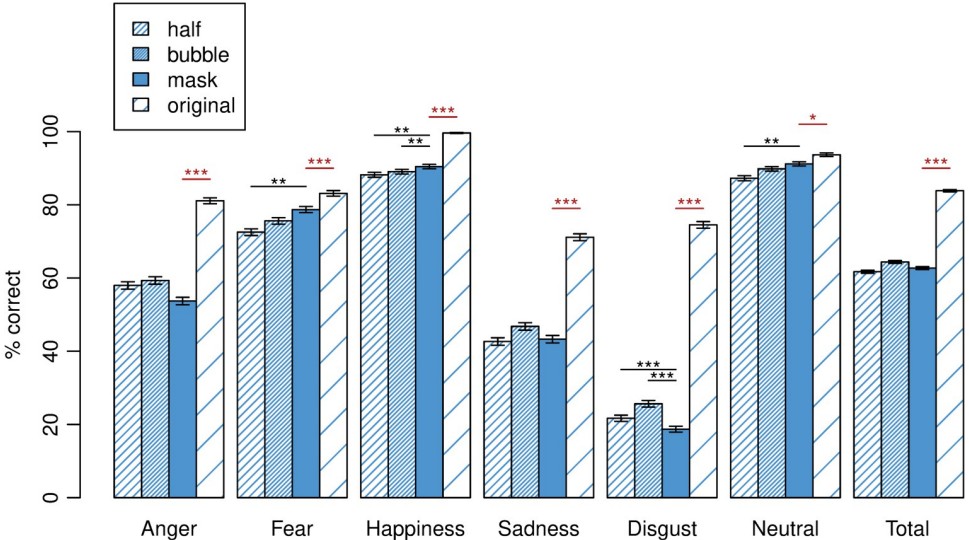

**Fig 2. Performance in the facial emotion recognition task for each type of occlusion.** Mean emotion recognition accuracy (percent correct) with error bars (standard error) for the separate emotions and across all (total) for each type of occlusion. *Mask vs. original* (n = 790), *half vs. bubble* (n = 395 and n = 388). Recognition accuracy of all emotions was significantly lower for 'half', 'bubble' and 'mask' stimuli compared to 'original' uncovered faces (all $p < .05$). Significance is indicated only for comparisons of 'mask' with 'original' (in red) and 'mask' with 'half' and 'bubble' (in black). *: $p < .05$, **: $p < .01$, ***: $p < .001$.

and type of occlusion, $F(15, 14154) = 199.77$, $p < .001$, $\eta_p^2 = .175$. To disentangle this significant interaction, we performed six ANOVAs, one for each emotion, revealing significant differences between the types of occlusion for all emotions: anger, $F(3, 2359) = 244.62$, $p < .001$, $\eta_p^2 = .237$, fear, $F(3, 2359) = 21.53$, $p < .001$, $\eta_p^2 = .027$, happiness, $F(3, 2359) = 102.48$, $p < .001$, $\eta_p^2 = .115$, sadness, $F(3, 2359) = 176.80$, $p < .001$, $\eta_p^2 = .184$, disgust, $F(3, 2359) = 1562.24$, $p < .001$, $\eta_p^2 = .665$, and neutral, $F(3, 2359) = 14.48$, $p < .001$, $\eta_p^2 = .018$. Effects of small to large size were obtained, with small effect sizes for neutral and fearful expressions and large effects for disgust, anger, sadness and happiness.

**Facial emotion recognition and face mask (*mask vs. original*).** Follow-up comparisons of emotion recognition for 'mask' vs. 'original' separately for each emotion showed that recognition accuracy was significantly lower in the mask condition compared to original uncovered faces for all emotions assessed, including angry ($p < .001$), fearful ($p = .001$), happy ($p < .001$), sad ($p < .001$), disgusted ($p < .001$) and neutral ($p = .016$) facial expressions. In the original condition, happy faces were recognised easiest, i.e., recognition scores were highest, closely followed by neutral expressions (see Fig 2). Angry expressions were recognised less frequently than neutral faces, followed by fearful, disgusted and, lastly, sad faces. Recognition accuracy of 'original' stimuli differed significantly between each emotion compared with each other emotion (all $p < .001$) apart from anger vs. fear ($p = .858$) and disgust vs. fear ($p = .160$). In the mask condition, happy and neutral expressions were recognised easiest as well, followed by fear, anger, sadness and, lastly, disgust. Recognition accuracy of 'mask' stimuli differed significantly between each emotion compared with each other emotion (all $p < .001$) apart from happiness vs. neutral ($p = .100$). S1 Fig 1 in S1 File additionally depicts the pattern of errors, i.e., misinterpretations of each emotion as another emotion (please see S1 File). Happy and neutral faces (both with and without mask) were rarely confused with any other emotion, while anger and sadness were frequently misinterpreted as disgust. Vice versa, masked faces showing

disgust were often mistaken for angry faces and both masked and unmasked faces depicting disgust were frequently misinterpreted as sad.

To check whether there is an effect of participant age on emotion recognition accuracy for masked and unmasked original faces in general as well as for each emotion, we performed linear regression analyses. Results from the analysis on the association of participant age with emotion recognition accuracy for masked faces revealed a significant effect of age, $F(1, 788) = 30.21$, $p < .001$, $R^2 = .04$, with lower recognition rates for faces with a mask for older participants. In contrast, the analysis on the association of participant age with recognition rates for unmasked original faces did not reveal a significant effect, $F(1, 788) = 1.38$, $p = .241$. See S1 Table 2 in S1 File for emotion-specific results.

**Facial emotion recognition and control occlusions (*half vs. bubble*).** Follow-up comparisons of emotion recognition for 'half' vs. 'original' and 'bubble' vs. 'original' separately for each emotion showed that recognition accuracy of all emotions was significantly lower in the half and bubble conditions compared to original uncovered faces (all $p < .01$), please see Fig 2.

Recognition of fearful and neutral faces was significantly less accurate in the half condition compared to the mask condition (both $p < .01$). For happy faces, recognition accuracy was significantly lower in the half condition and in the bubble condition compared to the mask condition (both $p < .01$), however, for disgusted expressions, recognition accuracy was significantly higher in the half and the bubble condition compared to the mask condition (both $p < .001$). There were no significant differences in recognition accuracy between the half and the bubble condition for any emotion (all $p > .05$). See S1 Table 3 in S1 File for details.

## Perception of threat

Fig 3 depicts the descriptive statistics of the perception of threat for type of occlusion. The ANOVA revealed a significant interaction between emotion and type of occlusion, $F(15, 14154) = 48.56$, $p < .001$, $\eta_p^2 = .049$. To disentangle the significant interaction, we performed six ANOVAs, one for each emotion, revealing significant differences between the types of occlusion for anger, $F(3, 2359) = 104.20$, $p < .001$, $\eta_p^2 = .117$, happiness, $F(3, 2359) = 18.57$, $p < .001$, $\eta_p^2 = .023$, sadness, $F(3, 2359) = 40.74$, $p < .001$, $\eta_p^2 = .049$, disgust, $F(3, 2359) = 97.58$,

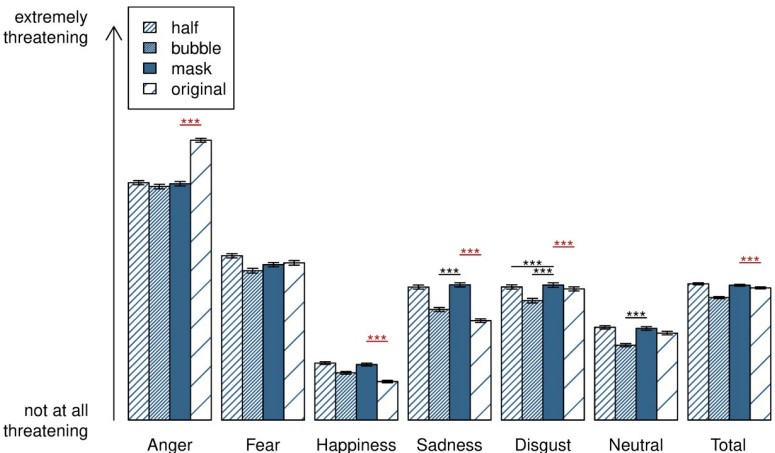

**Fig 3. Ratings of perception of threat for each type of occlusion.** Mean rating of threat on a Visual Analogue Scale (VAS) with error bars (standard error) for the separate emotions and across all (total) for each type of occlusion. *Mask vs. original* (n = 790), *half vs. bubble* (n = 395 and n = 388). Significance is indicated only for comparisons of 'mask' with 'original' (in red) and 'mask' with 'half' and 'bubble' (in black). ***: $p < .001$.

$p < .001$, $\eta_p^2 = .110$, and neutral, $F(3, 2359) = 6.73$, $p < .001$, $\eta_p^2 = .008$, but not for fear, $F(3, 2359) = 2.08$, $p = .100$. Effects were of small, medium and large size, with a small effect size for neutral and happy expressions, a medium effect size for sadness and large effects for anger and disgust.

**Perception of threat and face mask (*mask vs. original*).** Follow-up comparisons of perception of threat for 'mask' vs. 'original' separately for each emotion showed that faces with a happy, sad and disgusted expression were rated as significantly more threatening in the mask condition compared to the original condition (all $p < .001$). Contrarily, faces with an angry expression were rated as significantly less threatening in the mask condition compared to the original condition ($p < .001$). The same effects for mask vs. original were found for correctly recognised faces only: Please see S1 Table 4 in S1 File for the results of the same analysis as described above for rating of threat including only faces which were correctly recognised by participants.

In the original condition, angry faces were rated most threatening, followed by fearful, disgusted, sad, neutral and, lastly, happy expressions. In the mask condition, anger was rated most threatening, followed by disgust, fear, sadness, neutral and, lastly, happiness. Rated threat for each emotion was significantly different from rated threat for each other emotion, both for masked and unmasked faces (all $p < .05$).

To check whether there is an effect of participant age on the perception of threat for masked and unmasked original faces (in parallel to the effect of age on emotion recognition accuracy), we performed linear regression analyses. Results from the analysis on the association of participant age with rating of threat for masked faces revealed a significant effect of age, $F(1, 788) = 12.87$, $p < .001$, $R^2 = .02$, with older participants rating faces with a mask as less threatening. In contrast, the analysis on the association of participant age with perception of threat for unmasked original faces did not reveal a significant effect, $F(1, 788) = 1.74$, $p = .188$. See S1 Table 5 in S1 File for emotion-specific results.

**Perception of threat and control occlusions (*half vs. bubble*).** Follow-up comparisons of perception of threat for 'half' vs. 'original' separately for each emotion showed that faces with a happy and sad expression were rated as significantly more threatening in the half condition compared to the original condition (both $p < .001$). Comparisons of rating of threat for 'bubble' vs. 'original' showed that happy faces ($p < .01$) and neutral faces ($p < .05$) were rated as significantly more threatening in the bubble condition compared to the original condition. Contrarily, faces with an angry expression were rated as significantly less threatening in both the half and bubble condition compared to the original condition (both $p < .001$).

Sad and neutral faces were rated as significantly more threatening in the mask condition compared to the bubble condition (both $p < .001$) but not compared to the half condition (both $p > .05$), whereas faces with a disgusted expression were rated as significantly more threatening in the mask condition compared to the half and bubble condition (both $p < .001$). See S1 Table 6 in S1 File for details.

## Discussion

The aim of the present research was to determine whether the accuracy in recognising facial emotional expressions is affected by wearing face masks covering mouth and nose compared to faces without a mask (condition *mask vs. original*). Emotion recognition was assessed by an adapted version of a validated emotion recognition task, the VERT-K, investigating the recognition of the basic emotions anger, fear, happiness, sadness, disgust and neutral expressions. We expected emotion recognition to be less accurate for faces with a mask compared to faces without a mask. It was further investigated whether faces wearing a mask are perceived as

more threatening than faces without a mask. By implementing the condition *half vs. bubble*, we investigated whether emotion recognition performance and perception of threat for faces covered by a mask differs from recognition performance and perception of threat for faces where only the upper half is presented and for faces where a bubble occludes the lower facial regions. We found that emotion recognition was less accurate for faces wearing a mask covering mouth and nose (*mask vs. original*) and that this effect is not only due to occluding the lower part of the face in general, but that especially recognition of disgust seems to be impaired when wearing a face mask (*half vs. bubble*).

## The effect of face masks (*mask vs. original*)

The findings from our study suggest that emotion recognition is more difficult for faces wearing a face mask covering mouth and nose than for uncovered faces. In line with our hypothesis, we observed that emotion recognition was less accurate when faces were presented with a mask, for all emotions tested (anger, fear, happiness, sadness, disgust and neutral expressions). This finding is in accordance with previous research showing that emotion recognition is more difficult if the lower region of the face is occluded [17], though we can now extend this suggestion to also be true for (surgical) face masks, as other previous studies have suggested [34, 35, 38, 39]. We observed the largest decline in recognition performance due to the mask for faces expressing disgust. This supports previous reports indicating that different facial action units are important for recognition of different emotions, with the lower part of the face including the nose region being especially relevant for recognition of disgust [24, 25]. Happy expressions were recognised easiest both with and without face mask. This might be due to happiness being the only positive and therefore most distinct emotion tested in this study and generally being found to be recognised easily [10]. While happy expressions were rarely confused with any other emotion, anger and sadness were frequently misinterpreted as disgust, which is partly in line with previous research reporting systematic mistakes in emotion recognition [12]. Vice versa, disgust was often mistaken for sadness, both with and without face mask. In addition, masked faces showing disgust were frequently misinterpreted as angry.

We further found significant differences in the perception of threat dependent on face mask for faces showing anger, happiness, sadness and disgust. While angry faces were rated as less threatening, happy, sad and disgusted facial expressions were perceived as more threatening with mask than without. This could be explained by certain facial action units in the lower areas being occluded, leaving only the eyes and forehead visible: During expression of happiness, sadness and disgust, the eyes become smaller and wrinkles form around the eyes which might in turn be perceived as more threatening without the mouth yielding additional information, as narrowed eyes usually express anger [14]. This matches up with prior research that has suggested altered perceptions of closeness [34] and reduced trustworthiness [35] for faces occluded by a mask. Another possible explanation would be that these effects may be driven by a reduction in the emotional intensity perceived in faces occluded by a mask. This would result in e.g. masked happy faces seeming less happy to the observer than faces without mask (and therefore possibly more threatening) and angry faces with a mask appearing less angry compared to faces without mask (and therefore less threatening).

Furthermore, we found an association of participant age with emotion recognition accuracy for masked faces: Recognition rates were lower for older participants. We did not find an association of age with recognition accuracy for unmasked faces. Prior research investigating age differences in emotion recognition performance for fully visible faces report inconsistent emotion-specific effects: while emotion recognition has been found to be less accurate for fear [27, 28], anger [27–29] and sadness [28–30] in older participants (between 65–80 years of age),

improved recognition with older age (between 58–70 years) has been observed for happiness [30] and disgust [29]. Contrary to previous studies, we did not find a significant effect of age in emotion recognition for faces without a face mask, i.e., fully visible faces. This might be due to previous research investigating age differences by analysing age groups [27–30] instead of considering age as a continuous variable, as we did in the present study.

Similar to the findings on age differences in emotion recognition, we observed an association of participant age with perception of threat for masked faces but not for unmasked faces: Older participants rated masked faces as less threatening. This might be due to older participants having experienced face masks as more of a precautionary measure protecting from disease rather than as a health threat.

## The effect of other types of occlusion (*half vs. bubble*)

The findings from our study suggest that emotion recognition performance for faces covered by a mask differs from emotion recognition performance for faces occluded by something other than a face mask. Our findings show that there might be mask-specific effects on emotion recognition accuracy. We observed differences in recognition performance for fear, happiness and neutral expressions when comparing faces with a mask to faces where only the upper half was presented: Emotions were recognised more accurately when the face was wearing a mask. This finding may be due to the fact that, for half faces, a larger area of the face was occluded than for faces with a mask. In comparison to faces cut off beneath the eyes, the mask could have allowed for a vague idea about the outer shape of the lower face and whether the mouth might be open or closed. This might in turn have affected emotion-specific recognition rates.

However, we found a different pattern for disgust: Recognition rates for this emotion were higher for half faces and faces occluded by a bubble compared to masked faces. Especially this finding seems to be an effect specific to the face mask. It might be that it is possible to still surmise certain facial features that are particularly important for recognition of disgust in half faces and faces obscured by a bubble but not in faces with a mask. Previous research has shown that, for identification of certain emotions, some rather small action units have a higher importance for successful recognition than others [25]: For disgust, these particular action units seem to be grouped around the mouth. This would include raising the lip and plucking it, which would in turn also result in a wrinkled nose. Especially the wrinkled nose may still be discerned in half faces cut off at the onset of the nose and in faces occluded by a bubble only partly obscuring the cheeks but not in faces broadly covered by a surgical face mask.

Upon investigation of differences in perception of threat, when comparing masked faces to faces occluded by a bubble, we observed that sad and neutral expressions were rated as more threatening with a mask compared to when they were occluded by a bubble. Furthermore, we found that disgusted expressions were rated as more threatening when the face was wearing a mask than when it was occluded by a bubble as well as when only the upper half was presented. Especially with regard to disgust, it may be that certain facial areas covered by the face mask do not only negatively influence emotion recognition rates but thus render the facial expression highly equivocal which might then result in elevated perception of threat.

## Conclusion

The present study was able to confirm previous findings on impaired emotion recognition upon occlusion of the lower part of the face [17, 38, 39] by showing that emotion recognition performance was poorer for all types of occluded stimuli (mask, half faces, and obscuring bubbles) compared to faces that are fully visible. While recent studies mostly assessed emotion

recognition affected by face masks without control conditions [e.g. 35], we were able to report a specific effect of a certain kind of occlusion, namely the (surgical) face mask, especially for disgust. Not only did we observe mask-specific effects on emotion recognition performance, our results further revealed no significant difference in emotion recognition rates in the control condition using other types of occlusion. However, it should be noted that the bubble shape used in the present study may have been a poor approximation of the effect of a face mask and a rectangular shape may have been a more appropriate option to use as a control condition. The present study's results suggest that findings on emotion recognition in the context of occlusion should be interpreted with regard to the nature of the occlusion used in the respective study, as it might play a bigger role than previously assumed.

In comparison with recent reports on the effect of face masks on emotion recognition, our results once more illustrate that the general pattern of which facial action unit is important for recognition of which emotion at least partly depends on the measure [26]: While one study presented real images of people wearing masks for a limited duration (1000 ms) and conducted a control experiment with another type of occlusion [39], another study used stimuli with masks superimposed on existing images and presented them without time limit but did not control for general effects by another type of occlusion [38]. Furthermore, some studies only looked at a smaller number of emotions (e.g. only one negative and one positive emotion as well as a neutral expression, [33]), whereas other studies included distractor options [34]. In line with our findings, previous studies reported impaired emotion recognition performance for faces with face mask as well. However, the specific emotions used in prior research differ and findings concerning (decreased) recognition accuracy for specific emotions vary. This illustrates that results from studies investigating emotion recognition should be interpreted with regard to the methodological differences between studies.

Several methodological constraints have to be considered. As the stimulus set used in the current study was created several years ago, the quality of some stimuli could potentially be improved. Moreover, in the current paradigm, static images of faces were used, whereas, in a natural environment, emotional expressions would be observed within a certain context supported by gestures and movement. The intensity of facial expressions was high and it would be interesting to explore how facial expressions with lesser intensity will be recognised. For future studies, it is worth considering using video recordings of faces, as emotion recognition has been found to be more accurate for moving displays [17] and findings can be generalised more reliably. Furthermore, as was done in a previous study [39], the effect of other types of facial occlusion individuals tend to have more experience with, such as sunglasses or a niqab, should be considered.

Moreover, due to the nature of online-assessment, we were not able to collect data on response times in the current study but instead relied on measuring accuracy, i.e., the number of correct responses. Future studies assessing emotion recognition performance should consider collecting response times as well, which would allow for assessment of possible speed-accuracy trade-offs.

Another limitation of the present research is that, while the stimulus material used was validated for a German-speaking, i.e., possibly mostly Western, population, we did not assess the participants' ethnicity or cultural background. Previous studies have shown that recognition of basic emotions might actually not be culturally universal, as often assumed [44], but that facial expressions may be perceived and decoded differently dependent on the observer's culture and may especially differ between the Western and Eastern culture [45]. While the vast majority of the study samples in the present research was German-speaking (condition *mask vs. original*: 94%, condition *half vs. bubble*: 100%), we were not able to confirm that all participants were Caucasian or raised in a Western cultural background. Future studies should consider

assessing ethnicity, as assessment of the individual cultural background could inform about how accepting and familiar the observer is with face masks, as in East Asia, for instance, it has been customary to wear a mask in public for quite some time. For observers with an Asian background, faces with a mask may therefore not seem as threatening, bizarre or novel as they may seem to observers with a different cultural background. Future studies should therefore control for possible cultural effects and differences depending on ethnicity in connection with the stimulus material.

An interesting factor that was not investigated in detail in the current research but that has been shown to influence emotion recognition, is the sex of the observer. We decided to exclude the factor sex in our statistical analyses as, due to the nature of online sampling and the particular topic, we did not assess enough male participants to conduct powered analyses. Findings concerning sex differences in emotion recognition remain inconsistent, though most report higher recognition rates in women than men [3, 17, 46] (but see [27, 47] for no differences). It should be noted that between 73% to 80% of our participant sample were women, which might account for generally higher recognition rates than would be found in samples with a more equal sex ratio. Future studies on emotion recognition in the context of face masks might therefore be interested in considering the possible influence of sex and gender.

To conclude, we were able to show that wearing a face mask affects social interaction in terms of impaired emotion recognition accuracy and altered perception of threat. This might be especially relevant and possibly detrimental for individuals with certain mental disorders who show altered emotion recognition, for instance individuals with autism spectrum disorder, major depressive disorder or alcohol use disorder. For these persons, face masks may pose an additional obstacle during social interaction which is already impeded due to the nature of their mental disorder [48]. Future studies on the effect of face masks on emotion recognition and social interaction should therefore additionally focus on measures of social competence and assess whether psychopathology moderates the effects of face masks on emotion recognition. However, despite the limitations posed by mask wearing and the proposed effect on social interaction, there are many positive consequences of wearing face masks, such as touching one's face less often [49], which could in turn prevent the spread of infectious diseases.

## Supporting information

**S1 File. Supporting information.**
(PDF)

## Author Contributions

**Conceptualization:** Melina Grahlow, Birgit Derntl.

**Data curation:** Melina Grahlow, Claudia Ines Rupp.

**Formal analysis:** Melina Grahlow.

**Investigation:** Melina Grahlow.

**Methodology:** Melina Grahlow, Birgit Derntl.

**Project administration:** Melina Grahlow, Birgit Derntl.

**Resources:** Claudia Ines Rupp, Birgit Derntl.

**Software:** Melina Grahlow.

**Supervision:** Melina Grahlow, Birgit Derntl.

**Validation:** Melina Grahlow.

**Visualization:** Melina Grahlow.

**Writing – original draft:** Melina Grahlow, Birgit Derntl.

**Writing – review & editing:** Melina Grahlow, Claudia Ines Rupp, Birgit Derntl.

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
