## [Decision Letter · Decision Letter 0]

15 Jun 2021

PONE-D-21-12703

The impact of face masks on emotion recognition performance and perception of threat

PLOS ONE

Dear Dr. Grahlow,

Thank you for submitting your manuscript to PLOS ONE. After careful consideration, we feel that it has merit but does not fully meet PLOS ONE’s publication criteria as it currently stands. Therefore, we invite you to submit a revised version of the manuscript that addresses the points raised during the review process.

When revising their manuscript the authors are asked to pay special attention to the statistical/methodological comments of reviewer #2.

Moreover, it is particularly important that the paper addresses the macro issue of the relevance of this type of research (as suggested by reviewer #1). Please consider to briefly integrate the latest publications on the topic into the discussion of your results (e.g., Ruba and Pollak (2020, Plos One), Grundmann et al., (2021, Plos One), Calbi et al. (2021, Scientific Reports), Marini et al. (2021, Scientific Reports)).

We look forward to receiving your revised manuscript.

Kind regards,

Thomas Suslow, Ph.D.

Academic Editor

PLOS ONE

Journal Requirements:

2. Please change "female” or "male" to "woman” or "man" as appropriate, when used as a noun (see for instance https://apastyle.apa.org/style-grammar-guidelines/bias-free-language/gender).

4. "We note that Figure 1 and Figure 4 includes an image of a [patient / participant / in the study]. 

Reviewers' comments:

Reviewer's Responses to Questions

**Comments to the Author**

1. Is the manuscript technically sound, and do the data support the conclusions?

Reviewer #1: Yes

Reviewer #2: Yes

2. Has the statistical analysis been performed appropriately and rigorously? 

Reviewer #1: Yes

Reviewer #2: No

3. Have the authors made all data underlying the findings in their manuscript fully available?

Reviewer #1: Yes

Reviewer #2: Yes

4. Is the manuscript presented in an intelligible fashion and written in standard English?

Reviewer #1: Yes

Reviewer #2: Yes

5. Review Comments to the Author

Reviewer #1: The manuscript is concise and describes studies of emotion recognition of faces with and without a surgical mask, a highly relevant topic in the current context. Overall, the studies are well-designed (with one exception, see below) and the sample sizes are large. Overall, the study should make an important contribution to the emotion recognition literature. However, I have a number of constructive comments that should be addressed.

The manuscript covers important topics around emotion recognition, and does a good job tackling the “micro” issues in the introduction and discussion, but it does not address the “macro” issue of why this type of research is important and needed. It would help if the authors framed the present study around some bigger issues, and discuss how the current data might inform those issues. I can think of many relevant topics to frame this research, such as how mask effects might be relevant (harmful) to child development or interpersonal relations at the workplace, or have detrimental effects to persons with emotional processing abnormalities such as autism spectrum disorder or major depressive disorder. This is not a great problem, but more of a constructive comment that could improve the manuscript.

The introduction is concise and informative. However, there is a huge literature on the “Reading the Mind in the Eyes Test” (RMET) which has been used to assess emotion recognition. I would imagine that these data might inform the present study, as the stimuli used in the RMET only show the eyes and above region of the face.

Because we know that persons with certain mental disorders show altered emotion recognition (autism spectrum disorder, major depressive disorder, bipolar disorder, etc.), the paper would be stronger if the researchers could assess whether symptoms of a mental disorder (Beck Depression Inventory, for example) or social competence moderated the effects of mask wearing on emotion recognition.

Was reaction time collected in the studies? If so, was there any relationship between recognition errors and reaction time?

Also, please provide the exact phrasing for the forced-choice emotion recognition task. Were participants asked to identify the emotion as fast as possible, or was there no time pressure?

I would suggest that you be more clear on what a one sample t-test is actually testing, which is whether the difference score (mask minus no mask) is different than zero. I know this is implied, but it might help to be more explicit.

In addition, I would suggest that you provide a stronger rationale for your use of difference scores and how you structured the data analyses in study 1. I know that you are doing this because you are computing differences between masked faces and no mask faces at the level of the item, but do you really expect that there is so much inter-item variability that necessitates this approach? The issue with difference scores is the possibility that the level of error in the difference scores is larger than the level of error of the two base measures themselves. One could structure the analyses of recognition rates as a Mask X Emotion within-subject ANOVA (or mixed design ANOVA if you include between-subject factors such as sex of the participant), looking at main effects and the interaction. This would answer all of your research questions with a single omnibus test.

In the discussion, it is not clear what you mean by “the results for the specific emotions and their implications concerning the importance of facial action units differ and should be interpreted with regard to the methodological differences” (page 17). In particular, it is not clear how the “importance of facial action units” differs across studies. I would suggest expanding and presenting your point with greater clarity.

In the discussion, it is not clear why you chose to exclude sex in your statistical analyses. The authors refer to “online sampling”, but it is not clear why this is an issue. I suppose the authors concluded that there not enough male participants to conduct these analyses. Please explain this more clearly.

An important limitation of the study design, which comes out in the discussion of study 2, is that “bubble” shape seems to be a poor approximation of the effect of a mask. It would be more appropriate to use a rectangular shape that approximates the shape of a mask. I would suggest adding this as a limitation.

The paper is well written. I only have a few suggestions.

Page 7: in the sentence “To assess whether faces wearing a mask are perceived as more threatening than faces without a mask, a Visual Analogue Scale (VAS) was applied”, the verb applied is not appropriate. I would replace it with “used” or “selected”.

Page 7: “To test whether emotion recognition performance differs dependent on face mask,…” You might want to phrase this as “..depending on the presence of a face mask,…”

Page 10: “While happy expressions were rarely confused with any other emotion, especially anger and sadness were frequently misinterpreted as disgust, which is partly in line with previous research reporting systematic mistakes in emotion recognition (8). Vice versa, disgust was often mistaken for sadness, both with and without face mask.” This sentence could be better phrased (removing “especially” would help). Also, it seems that you are referring to data from your study, but I am not sure where these data are presented. In the supplemental tables? Please clarify, and indicate to the reader (this could be done in the results section) where these data are.

Reviewer #2: This manuscript investigated the impact of face masks on emotional face categorisation. In study 1, categorisation accuracy and threat intensity were measured for original images, and those with faces masks superimposed. In study 2, categorisation accuracy and threat intensity were measured for face halves and faces partially covered by an ellipse. The paper is generally well-written and clear. I have a few comments that the authors might find helpful. Most are relatively minor, but I do query the two-study approach, given that all the data from the first study are analysed again in the second (see below).

It isn’t clear why difference scores are calculated in study 1, but not in study 2. It can be quite difficult to tell if difference scores are hiding important aspects of the data. Given that study 1 is effectively covered again in study 2, it might make more sense just to present all data together in one study, with one large mixed ANOVA (i.e. remove Study 1).

Is it possible that the threat-related effects are driven by a reduction in the emotional intensity perceived in masked faces? I.e. Happy faces become less happy (and therefore more threatening), and angry faces become less angry (and therefore less threatening)?

It would be helpful to have a comment on the size of the effects found – were these generally large effects, or were they relatively small?

P. 3, 64: bottom face half is stated as “superior“ for anger recognition, but in the next sentence, the top face half is described as “especially important” for anger recognition. It’s fine if both are important, but both can’t be the most important.

P.4, 94: The description of this study is incorrect. There were many differences between the effects of sunglasses and masks on the categorisation of emotion (there were fewer in the identity matching tasks).

P5, 117: Odd phrasing in this final paragraph –“studies online achieving our overall aim of the study was to..”

P5, 123: Specify what is meant by “more difficult” here – typically it’s meant to mean more incorrect responses and longer response times, whereas only accuracy is measured in the current experiment. It would be helpful to have acknowledgement of this and the fact that speed-accuracy trade-offs can exist, so accuracy data alone may not tell the whole story.

Sample: what was the reasoning behind including older adults in the sample? Older adults may have differences in emotional face recognition compared to younger adults.

P.7, 169: Clarify whether participants had an unlimited time to view each image, and whether the response options were presented at the same time as the image.

P.9, 250: If I’m reading it correctly, it seems odd that there are no differences in rated threat between emotions, given that both happy and angry expressions are presented. Is this threat measure sensitive?

P 10, 271-275: It’s implied that all expressions are affected by masks, but that’s not what has been found in the results (where happy, fearful and neutral faces were not significantly affected).

P 12, 360: State whether between or within participant variables were entered into the ANOVA.

P 17, 522: I don’t think we should assume that German speaking individuals are white. I would be more comfortable if this sentence was removed from the manuscript.

6. PLOS authors have the option to publish the peer review history of their article (what does this mean?). If published, this will include your full peer review and any attached files.

Reviewer #1: **Yes: **Mark A Ellenbogen

Reviewer #2: No

---

## [Author Response · Author response to Decision Letter 0]

28 Jul 2021

We would like to thank both reviewers and the Academic Editor for their constructive and insightful comments that helped improving our manuscript. Please find our response in detail below. Any changes within the manuscript are highlighted in yellow.

Academic Editor

Thank you for submitting your manuscript to PLOS ONE. After careful consideration, we feel that it has merit but does not fully meet PLOS ONE’s publication criteria as it currently stands. Therefore, we invite you to submit a revised version of the manuscript that addresses the points raised during the review process.

When revising their manuscript, the authors are asked to pay special attention to the statistical/methodological comments of reviewer #2. Moreover, it is particularly important that the paper addresses the macro issue of the relevance of this type of research (as suggested by reviewer #1). Please consider to briefly integrate the latest publications on the topic into the discussion of your results (e.g., Ruba and Pollak (2020, Plos One), Grundmann et al., (2021, Plos One), Calbi et al. (2021, Scientific Reports), Marini et al. (2021, Scientific Reports)).

Response: Thank you very much for the opportunity to revise our manuscript and to resubmit. We appreciate the acknowledgement of our work so far. Thank you for pointing out the latest publications on the topic. We have integrated and discussed the latest findings in the context of our manuscript. Further, we have paid special attention to address the macro issue of the relevance of our research and made changes according to the methodological/statistical comments of reviewer #2.

Journal Requirements

1) Please ensure that your manuscript meets PLOS ONE's style requirements, including those for file naming. The PLOS ONE style templates can be found at […].

Response: Thank you for this comment and for pointing out where the style templates can be found. To the best of our knowledge, we have now ensured that our manuscript meets PLOS ONE’s style and file naming requirements.

2) Please change "female” or "male" to "woman” or "man" as appropriate, when used as a noun (see for instance […]).

Response: Thank you for this comment. We have made changes accordingly throughout the manuscript.

3) We note that you have stated that you will provide repository information for your data at acceptance. Should your manuscript be accepted for publication, we will hold it until you provide the relevant accession numbers or DOIs necessary to access your data. If you wish to make changes to your Data Availability statement, please describe these changes in your cover letter and we will update your Data Availability statement to reflect the information you provide.

Response: Thank you for pointing out the missing repository information for our data. Please find the relevant DOI necessary to access our data in the Open Science Framework here: https://doi.org/10.17605/OSF.IO/JY2ES. We have also included this information in our cover letter.

4) We note that Figure 1 and Figure 4 includes an image of a [patient / participant / in the study]. As per the PLOS ONE policy […] on papers that include identifying, or potentially identifying, information, the individual(s) or parent(s)/guardian(s) must be informed of the terms of the PLOS open-access (CC-BY) license and provide specific permission for publication of these details under the terms of this license. Please download the Consent Form for Publication in a PLOS Journal […]. The signed consent form should not be submitted with the manuscript but should be securely filed in the individual's case notes. Please amend the methods section and ethics statement of the manuscript to explicitly state that the patient/participant has provided consent for publication: “The individual in this manuscript has given written informed consent (as outlined in PLOS consent form) to publish these case details”. If you are unable to obtain consent from the subject of the photograph, you will need to remove the figure and any other textual identifying information or case descriptions for this individual.

Response: Thanks for this information. Please note that the individuals' faces depicted in former Figure 1 and Figure 4 (now both in Figure 1) are part of the stimulus set used in the experiment - the Vienna Emotion Recognition Tasks (VERT-K; Derntl et al., 2008) - which in turn goes back to a validated and published stimulus set of facial expressions by Gur et al. (2002). This was stated in the manuscript in the methods section, however, we have now also included this information in the figure description. Furthermore, we have obtained permission by the author, Ruben C. Gur, to publish the respective images under a CC BY open access license (please also see the separate document attached labelled ‘Gur_Permission’).

“Fig 1. Exemplary stimuli of the adapted Vienna Emotion Recognition Tasks (VERT-K). 

A female poser showing an angry expression and a male poser showing a happy expression. Exemplary stimuli as used in experiment 1 (‘original’ and ‘mask’) and control stimuli used in experiment 2a (‘half’) and experiment 2b (‘bubble’). Images extracted and modified from (6).”

Reviewer 1

The manuscript is concise and describes studies of emotion recognition of faces with and without a surgical mask, a highly relevant topic in the current context. Overall, the studies are well-designed (with one exception, see below) and the sample sizes are large. Overall, the study should make an important contribution to the emotion recognition literature. However, I have a number of constructive comments that should be addressed.

Response: Thank you for acknowledging the timeliness of our study and for your opinion on the novelty of our research.

1) The manuscript covers important topics around emotion recognition, and does a good job tackling the “micro” issues in the introduction and discussion, but it does not address the “macro” issue of why this type of research is important and needed. It would help if the authors framed the present study around some bigger issues and discuss how the current data might inform those issues. I can think of many relevant topics to frame this research, such as how mask effects might be relevant (harmful) to child development or interpersonal relations at the workplace or have detrimental effects to persons with emotional processing abnormalities such as autism spectrum disorder or major depressive disorder. This is not a great problem, but more of a constructive comment that could improve the manuscript.

Response: We thank the reviewer for this very valuable advice. It showed us that our manuscript needed to frame the research topic of emotion recognition of faces with and without a face mask in a broader context and to emphasise the possibly detrimental effects of face masks on social interaction for specific groups of individuals. Thank you for suggesting some bigger issues in which face masks might have detrimental effects. Following your advice, we have addressed the macro issue of the relevance of this type of research more elaborately in the revised introduction section (please see page 3):

“Emotions have a regulative social function and constitute an important factor in appropriate interaction between individuals: Emotional expressions instigate informative processes that influence individuals’ behaviour during social interaction (2). For instance, the happy facial expression of your counterpart during a conversation will most likely encourage you to keep telling that story, whereas an angry expression might let you quickly change the subject. The success of our social interaction depends on the accuracy of emotion recognition, as different emotions convey different information by facial cues, such as the movement of certain muscles in the eye, nose and mouth regions (3). Prototypical (surgical) face masks as worn during a pandemic to reduce the risk of infection, may reduce the likelihood of accurate emotion recognition. These face masks typically cover mouth and nose and leave only the upper areas of the face visible, which might undermine the success of our social interactions by reducing the number of available facial cues. The hindering effect of face masks on accurate emotion recognition may be especially relevant in interpersonal relations at the workplace where appropriate social interaction is expected. Furthermore, concerns on how face masks might (harmfully) impact child development have been voiced and it has been suggested that certain communication techniques might be important to ensure effective communication with children further on (4). Effects of face masks are also relevant in the context of mental disorders: Individuals with emotional processing alterations such as e.g. individuals with autism spectrum disorder have been shown to employ perceptual strategies that are not optimal for face-processing, such as focussing mostly on the mouth region instead of directing their gaze to the eye region as well (5). It becomes clear that not only specific groups of individuals such as those with a mental disorder but each and every one of us depends on successful social interaction, which is why it is important to investigate the possibly detrimental effect of face masks on emotion recognition performance across the general population.”

2) The introduction is concise and informative. However, there is a huge literature on the “Reading the Mind in the Eyes Test” (RMET) which has been used to assess emotion recognition. I would imagine that these data might inform the present study, as the stimuli used in the RMET only show the eyes and above region of the face.

Response: Thank you for this important comment and for the excellent literature suggestion. We have studied the literature carefully and have included findings from the “Reading the Mind in the Eyes Test” in the introduction (please see page 4):

“Several studies have stressed the importance of facial areas in emotion recognition. During free inspection of faces, observers tend to focus mostly on the eye region (15). Studies using the ‘Reading the Mind in the Eyes Test’ (16) show that the eye region is informative of the mental state, i.e. the feelings and thoughts, of a person and may thus pose an important facial area for emotion recognition. However, previous findings are inconclusive concerning which facial areas might be the most important for correct recognition of specific emotions: While emotion recognition is most accurate if participants are presented with full face displays, it was found to be less accurate when only the lower region of the face can be seen and the least accurate if participants only see the upper region of the face (17).”

3) Because we know that persons with certain mental disorders show altered emotion recognition (autism spectrum disorder, major depressive disorder, bipolar disorder, etc.), the paper would be stronger if the researchers could assess whether symptoms of a mental disorder (Beck Depression Inventory, for example) or social competence moderated the effects of mask wearing on emotion recognition.

Response: Thank you for raising this important point. We agree that the issue raised by the reviewer is highly relevant and that the effect of facial masks on emotion recognition in individuals with certain mental disorders should be investigated in future studies. We had considered assessing symptoms of mental disorders while designing the study, however, due to economic validity, we decided to keep the questionnaire as short as possible and to forego assessing mental health for the time being. At the moment, we are conducting a study on the effect of Covid-19 on university students’ mental health (including a research question on face masks and emotion recognition with respect to possible mental disorders) in close collaboration with partners in Innsbruck (Austria, co-author Claudia I. Rupp) and Aachen (Germany). We have further included the point raised by the reviewer as a suggestion for future research in the revised discussion (please see page 24):

“To conclude, we were able to show that wearing a face mask affects social interaction in terms of impaired emotion recognition accuracy and altered perception of threat. This might be especially relevant and possibly detrimental for individuals with certain mental disorders who show altered emotion recognition, for instance individuals with autism spectrum disorder, major depressive disorder or alcohol use disorder. For these persons, face masks may pose an additional obstacle during social interaction which is already impeded due to the nature of their mental disorder (48). Future studies on the effect of face masks on emotion recognition and social interaction should therefore additionally focus on measures of social competence and assess whether psychopathology moderates the effects of face masks on emotion recognition. However, we may want to accept limitations posed by mask wearing to a certain extent, as there are many positive consequences of wearing face masks, such as touching one’s face less often (49), which could in turn prevent the spread of infectious diseases.”

4) Was reaction time collected in the studies? If so, was there any relationship between recognition errors and reaction time?

Also, please provide the exact phrasing for the forced-choice emotion recognition task. Were participants asked to identify the emotion as fast as possible, or was there no time pressure?

Response: Thank you for your interest in the outcomes of the study. Due to the nature of the online tool we used, it was not possible to collect reaction times as this would have been highly dependent on the stability of the participants’ internet connection. Instead, we presented one image as well as the response options and Visual Analogue Scale on one page at a time. There was no time pressure for participants to identify the emotion depicted. The exact phrasing for the forced-choice emotion recognition task was the following:

„On the following pages, you will see images of different faces.

Please take a close look at every face and decide which emotion is expressed by each person. Once you have decided, please tick the corresponding word.

If the person does not show any emotion, please click 'neutral'.

In addition, you will be asked to rate each face on a scale from not threatening at all to extremely threatening. You can move the slider on the scale to the right and to the left.“

5) I would suggest that you be more clear on what a one sample t-test is actually testing, which is whether the difference score (mask minus no mask) is different than zero. I know this is implied, but it might help to be more explicit.

Response: Thank you for this suggestion. In our revised manuscript, we followed both reviewers’ advice to reconsider the use of difference scores (please see reviewer 1, section 6 and reviewer 2, section 1). We now present all data in one single study making use of a single omnibus ANOVA instead. As we have omitted the analysis of difference scores using one sample t-tests, there is no further need for changes in our manuscript in this regard.

6) In addition, I would suggest that you provide a stronger rationale for your use of difference scores and how you structured the data analyses in study 1. I know that you are doing this because you are computing differences between masked faces and no mask faces at the level of the item, but do you really expect that there is so much inter-item variability that necessitates this approach? The issue with difference scores is the possibility that the level of error in the difference scores is larger than the level of error of the two base measures themselves. One could structure the analyses of recognition rates as a Mask X Emotion within-subject ANOVA (or mixed design ANOVA if you include between-subject factors such as sex of the participant), looking at main effects and the interaction. This would answer all of your research questions with a single omnibus test.

Response: We thank the reviewer for raising this issue, which showed us that the use of difference scores is prone to statistical errors and that the rationale for our use of difference scores was not strong enough. We fully agree with the reviewer that structuring the analyses of recognition rates as a mask x emotion within-subject ANOVA would have been the analysis of choice. However, our data did not allow for a within-subject analysis, as not all participants saw all faces both with and without a face mask. Due to economic validity and time considerations, and to ensure viability, the experiment in Study 1 was split into two versions so that each poser was only seen once in each version (if poser A was wearing a face mask in version A, they were presented without a face mask in version B). Participants were quasi-randomly assigned to one of the two versions. Despite the limitation of not being able to use within-subject analyses, we have assessed a very large sample size, which allows us to draw meaningful conclusions from the data nonetheless.

To address the issue raised by the reviewer, and following further advice from reviewer 2 (see reviewer 2, section 1), we have omitted the use of difference scores and now jointly report results from former Study 1 and Study 2 as one single study consisting of several experiments, making use of a single omnibus ANOVA, with emotion as within-subject and type of occlusion as between-subjects factors. Please see our revised statistical analyses section (page 11):

“Statistical analyses were performed using SPSS 25 (IBM SPSS Statistics) with alpha set to .05. Despite minor deviations from normality (visual inspection and Kolmogorov-Smirnoff tests), we relied on parametric tests including analysis of variance (ANOVA) as these are adequate due to the reasonably large sample sizes (40).

Separately for experiment 1, 2a and 2b, the number of correct responses was calculated for each emotion, resulting in a mean score of emotion recognition accuracy (percent correct) for each participant per emotion for faces with and without mask (experiment 1), for half faces (experiment 2a) and for faces with a bubble obscuring mouth and nose (experiment 2b).

The current study aimed to investigate whether emotion recognition performance differs depending on the presence of a face mask (experiment 1, ‘mask’ and ‘original’) or another type of occlusion of the lower facial areas (‘half’ in experiment 2a and ‘bubble’ in experiment 2b). For this purpose, we relied on a 6 x 4 repeated-measures analysis of variance (ANOVA) with emotion (anger, fear, happiness, sadness, disgust and neutral) as within-subject factor and type of occlusion (half, bubble, mask and original) as between-subjects factor. For significant interactions, we performed separate emotion-specific ANOVAs.

To analyse whether participants perceive faces wearing a mask as more threatening than faces without a mask and to test for differences in perception of threat depending on the type of occlusion, individual ratings on the VAS were averaged, resulting in a mean score of perception of threat for each participant per emotion for each type of occlusion. These scores were analysed using a 6 x 4 ANOVA with the same factors as stated above. For significant interactions, separate ANOVAs were performed for each emotion.

For all ANOVAs, where sphericity was violated, Greenhouse-Geisser adjusted values are reported. Post-hoc tests with Bonferroni correction were calculated for significant results. We further report estimates of effect size for significant results using partial-eta-squared.”

7) In the discussion, it is not clear what you mean by “the results for the specific emotions and their implications concerning the importance of facial action units differ and should be interpreted with regard to the methodological differences” (page 17). In particular, it is not clear how the “importance of facial action units” differs across studies. I would suggest expanding and presenting your point with greater clarity.

Response: Thank you for pointing out that the phrasing in the manuscript was not clear enough to bring across the point we wanted to make. We have rephrased this section in the discussion (please see page 22):

“In line with our findings, previous studies reported impaired emotion recognition performance for faces with face mask as well. However, the specific emotions used in prior research differ and findings concerning (decreased) recognition accuracy for specific emotions vary. This illustrates that results from studies investigating emotion recognition should be interpreted with regard to the methodological differences between studies.”

8) In the discussion, it is not clear why you chose to exclude sex in your statistical analyses. The authors refer to “online sampling”, but it is not clear why this is an issue. I suppose the authors concluded that there not enough male participants to conduct these analyses. Please explain this more clearly.

Response: We agree with the reviewer that this is important information that was lacking in the manuscript. Indeed, we concluded that there were not enough male participants to conduct analyses including the sex of participants. We have included a statement accordingly in the revised discussion (please see page 24):

“An interesting factor that was not investigated in detail in the current research but that has been shown to influence emotion recognition, is the sex of the observer. We decided to exclude the factor sex in our statistical analyses as, due to the nature of online sampling and the particular topic, we did not assess enough male participants to conduct powered analyses.”

9) An important limitation of the study design, which comes out in the discussion of study 2, is that “bubble” shape seems to be a poor approximation of the effect of a mask. It would be more appropriate to use a rectangular shape that approximates the shape of a mask. I would suggest adding this as a limitation.

Response: Thank you for pointing out the limitation that the bubble shape does not seem to approximate the shape of a face mask sufficiently. We chose the bubble shape instead of a rectangular shape to create a control stimulus that is effectually distinct from the other control condition using half faces that were cut off just beneath the eyes, resulting in a rectangular type of occlusion. We have included and discussed this issue as a limitation in the discussion (please see page 21):

“Not only did we observe mask-specific effects on emotion recognition performance, our results further revealed no significant difference in emotion recognition rates between the two control experiments using other types of occlusion (experiment 2a and 2b). However, it should be noted that the bubble shape used in the present study may have been a poor approximation of the effect of a face mask and a rectangular shape may have been a more appropriate option to use as a control condition.”

10) The paper is well written. I only have a few suggestions.

Page 7: in the sentence “To assess whether faces wearing a mask are perceived as more threatening than faces without a mask, a Visual Analogue Scale (VAS) was applied”, the verb applied is not appropriate. I would replace it with “used” or “selected”.

Page 7: “To test whether emotion recognition performance differs dependent on face mask,...” You might want to phrase this as “..depending on the presence of a face mask,...”

Response: Thank you for the suggestions concerning the use of words and rephrasing. We appreciate the opportunity to improve our style of writing and have made changes in the methods section accordingly, please see page 10 and page 11:

Methods - page 10:

“To assess whether faces wearing a mask are perceived as more threatening than faces without a mask, a Visual Analogue Scale (VAS) was used. Participants were asked to indicate how they perceived each face on a scale ranging from 0 (not threatening at all) to 100 (extremely threatening).”

Methods - page 11:

“The current study aimed to investigate whether emotion recognition performance differs depending on the presence of a face mask (experiment 1, ‘mask’ and ‘original’) or another type of occlusion of the lower facial areas (‘half’ in experiment 2a and ‘bubble’ in experiment 2b).”

11) Page 10: “While happy expressions were rarely confused with any other emotion, especially anger and sadness were frequently misinterpreted as disgust, which is partly in line with previous research reporting systematic mistakes in emotion recognition (8). Vice versa, disgust was often mistaken for sadness, both with and without face mask.” This sentence could be better phrased (removing “especially” would help). Also, it seems that you are referring to data from your study, but I am not sure where these data are presented. In the supplemental tables? Please clarify, and indicate to the reader (this could be done in the results section) where these data are.

Response: We thank the reviewer for this hint. We are indeed referring to data from our study, which can be found in Figure S1 in the supplement, which depicts the pattern of errors, i.e. misinterpretations of each emotion as another emotion. This was indicated to the reader in the results section, just after presentation of the emotion recognition performance for mask vs. no mask. We have now additionally included a short description of the pattern of errors in the results section, please see page 13. Further, we have rephrased the respective sentence in the discussion, please see page 18:

Results – page 13:

“In the mask condition, happy and neutral expressions were recognised easiest as well, followed by fear, anger, sadness and, lastly, disgust. Recognition accuracy of ‘mask’ stimuli differed significantly between each emotion compared with each other emotion (all p < .001) apart from happiness vs. neutral (p = .100). S1 Fig additionally depicts the pattern of errors, i.e. misinterpretations of each emotion as another emotion (please see supplement). Happy and neutral faces (both with and without mask) were rarely confused with any other emotion, while anger and sadness were frequently misinterpreted as disgust. Vice versa, masked faces showing disgust were often mistaken for angry faces and both masked and unmasked faces depicting disgust were frequently misinterpreted as sad.”

Discussion – page 18:

“While happy expressions were rarely confused with any other emotion, anger and sadness were frequently misinterpreted as disgust, which is partly in line with previous research reporting systematic mistakes in emotion recognition (12). Vice versa, disgust was often mistaken for sadness, both with and without face mask. In addition, masked faces showing disgust were frequently misinterpreted as angry.”

Reviewer 2

This manuscript investigated the impact of face masks on emotional face categorisation. In study 1, categorisation accuracy and threat intensity were measured for original images, and those with face masks superimposed. In study 2, categorisation accuracy and threat intensity were measured for face halves and faces partially covered by an ellipse. The paper is generally well-written and clear. I have a few comments that the authors might find helpful. Most are relatively minor, but I do query the two-study approach, given that all the data from the first study are analysed again in the second (see below).

1) It isn’t clear why difference scores are calculated in study 1, but not in study 2. It can be quite difficult to tell if difference scores are hiding important aspects of the data. Given that study 1 is effectively covered again in study 2, it might make more sense just to present all data together in one study, with one large mixed ANOVA (i.e. remove Study 1).

Response: We thank the reviewer for raising this important point. In Study 1, the data allowed for calculation of difference scores, as participants were presented faces with mask as well as faces without mask. For reasons of economic validity, in Study 2, participants were presented control stimuli only, i.e. ‘half’ or ‘bubble’ stimuli, without seeing un-masked faces once more. Hence, the calculation of difference scores within the same sample was only possible in Study 1 but not in Study 2.

We fully agree with the reviewer that presenting all data together in one study with one large ANOVA is more adequate, especially as we had intended to conduct the control experiments ab initio. Following the reviewer’s advice, we have removed Study 1 and now present all data together in one study, with one single omnibus ANOVA. Please see our revised statistical analyses section (page 11):

“Statistical analyses were performed using SPSS 25 (IBM SPSS Statistics) with alpha set to .05. Despite minor deviations from normality (visual inspection and Kolmogorov-Smirnoff tests), we relied on parametric tests including analysis of variance (ANOVA) as these are adequate due to the reasonably large sample sizes (40).

Separately for experiment 1, 2a and 2b, the number of correct responses was calculated for each emotion, resulting in a mean score of emotion recognition accuracy (percent correct) for each participant per emotion for faces with and without mask (experiment 1), for half faces (experiment 2a) and for faces with a bubble obscuring mouth and nose (experiment 2b).

The current study aimed to investigate whether emotion recognition performance differs depending on the presence of a face mask (experiment 1, ‘mask’ and ‘original’) or another type of occlusion of the lower facial areas (‘half’ in experiment 2a and ‘bubble’ in experiment 2b). For this purpose, we relied on a 6 x 4 repeated-measures analysis of variance (ANOVA) with emotion (anger, fear, happiness, sadness, disgust and neutral) as within-subject factor and type of occlusion (half, bubble, mask and original) as between-subjects factor. For significant interactions, we performed separate emotion-specific ANOVAs.

To analyse whether participants perceive faces wearing a mask as more threatening than faces without a mask and to test for differences in perception of threat depending on the type of occlusion, individual ratings on the VAS were averaged, resulting in a mean score of perception of threat for each participant per emotion for each type of occlusion. These scores were analysed using a 6 x 4 ANOVA with the same factors as stated above. For significant interactions, separate ANOVAs were performed for each emotion.

For all ANOVAs, where sphericity was violated, Greenhouse-Geisser adjusted values are reported. Post-hoc tests with Bonferroni correction were calculated for significant results. We further report estimates of effect size for significant results using partial-eta-squared.”

2) Is it possible that the threat-related effects are driven by a reduction in the emotional intensity perceived in masked faces? I.e. Happy faces become less happy (and therefore more threatening), and angry faces become less angry (and therefore less threatening)?

Response: We thank the reviewer for the interest in the outcomes of the study and for suggesting further interpretation of the data. We agree that threat-related effects may also partly be driven by a mask-induced reduction of the emotional intensity of facial expressions and have included this point in the discussion (please see page 18):

“We further found significant differences in the perception of threat dependent on face mask for faces showing anger, happiness, sadness and disgust. While angry faces were rated as less threatening, happy, sad and disgusted facial expressions were perceived as more threatening with mask than without. This could be explained by certain facial action units in the lower areas being occluded, leaving only the eyes and forehead visible: During expression of happiness, sadness and disgust, the eyes become smaller and wrinkles form around the eyes which might in turn be perceived as more threatening without the mouth yielding additional information, as narrowed eyes usually express anger (14). This matches up with prior research that has suggested altered perceptions of closeness (30) and reduced trustworthiness (31) for faces occluded by a mask. Another possible explanation would be that these effects may be driven by a reduction in the emotional intensity perceived in faces occluded by a mask. This would result in e.g. masked happy faces seeming less happy to the observer than faces without mask (and therefore possibly more threatening) and angry faces with a mask appearing less angry compared to faces without mask (and therefore less threatening).”

3) It would be helpful to have a comment on the size of the effects found – were these generally large effects, or were they relatively small?

Response: Thank you for pointing out the missing information. The effects of the type of occlusion on emotion recognition performance and perception of threat ranged from small to large effects, which is comparable to effect sizes found in previous studies (e.g. Carbon, 2020). We included comments on the size of effects found in the results section, please see page 12 and page 15:

Facial emotion recognition – page 12:

“To disentangle this significant interaction, we performed six ANOVAs, one for each emotion, revealing significant differences between the types of occlusion for all emotions: anger, F(3, 2359) = 244.62, p < .001, η_p^2 = .237, fear, F(3, 2359) = 21.53, p < .001, η_p^2 = .027, happiness, F(3, 2359) = 102.48, p < .001, η_p^2 = .115, sadness, F(3, 2359) = 176.80, p < .001, η_p^2 = .184, disgust, F(3, 2359) = 1562.24, p < .001, η_p^2 = .665, and neutral, F(3, 2359) = 14.48, p < .001, η_p^2 = .018. Effects of small to large size were obtained, with small effect sizes for neutral and fearful expressions and large effects for disgust, anger, sadness and happiness.”

Perception of threat – page 15:

“To disentangle the significant interaction, we performed six ANOVAs, one for each emotion, revealing significant differences between the types of occlusion for anger, F(3, 2359) = 104.20, p < .001, η_p^2 = .117, happiness, F(3, 2359) = 18.57, p < .001, η_p^2 = .023, sadness, F(3, 2359) = 40.74, p < .001, η_p^2 = .049, disgust, F(3, 2359) = 97.58, p < .001, η_p^2 = .110, and neutral, F(3, 2359) = 6.73, p < .001, η_p^2 = .008, but not for fear, F(3, 2359) = 2.08, p = .100. Effects were of small, medium and large size, with a small effect size for neutral and happy expressions, a medium effect size for sadness and large effects for anger and disgust.”

4) P 3, 64: bottom face half is stated as “superior“ for anger recognition, but in the next sentence, the top face half is described as “especially important” for anger recognition. It’s fine if both are important, but both can’t be the most important.

Response: Thanks for spotting this error. We have rephrased the section accordingly, please see page 5:

“The lower region of the face has been found to be superior in recognition of happiness, while the upper region of the face seems to be important for recognition of fearful and sad expressions (19–21). For recognition of anger, results remain inconsistent with some studies stressing the importance of the eye region (22) while others find lower recognition accuracy for anger upon occlusion of the mouth (23). For disgust, it is neither solely the lower nor the upper region of the face but more the nose and cheek region that seems to yield the most important information for recognition (24).”

5) P 4, 94: The description of this study is incorrect. There were many differences between the effects of sunglasses and masks on the categorisation of emotion (there were fewer in the identity matching tasks).

Response: We thank the reviewer for making us aware that our description of the outcomes of the study by Noyes et al. (2019) was incorrect. We have corrected and rephrased the description accordingly, please see page 6:

“Another study investigated the effect of face masks and sunglasses (an occlusion individuals tend to have more experience with) on emotion recognition and found that, while recognition performance was reduced in both cases, accuracy was even lower in the mask condition (35). Further research comparing the effect of sunglasses and face masks on emotion recognition performance in children found that children were still able to make accurate inferences about the emotional expressions shown, even when the faces were partly covered by sunglasses or a mask (36). While a small effect of face coverings on emotion recognition accuracy was shown, the sunglasses and mask conditions did not differ significantly from each other. This remains to be investigated more closely by inclusion of further mask-related control conditions, as sunglasses normally only cover the eye region; face masks, however, cover the lower half of the face, thus both occlusion types potentially hindering recognition of different facial emotional expressions.”

6) P 5, 117: Odd phrasing in this final paragraph –“studies online achieving our overall aim of the study was to..”

Response: Thanks for pointing out this odd phrasing. We have made changes accordingly, see page 7:

“Using convenience sampling, we recruited large samples of individuals who completed the tasks online, achieving our overall aim of investigating the influence of a (surgical) face mask on emotion recognition performance and perception of threat.”

7) P 5, 123: Specify what is meant by “more difficult” here – typically it’s meant to mean more incorrect responses and longer response times, whereas only accuracy is measured in the current experiment. It would be helpful to have acknowledgement of this and the fact that speed-accuracy trade-offs can exist, so accuracy data alone may not tell the whole story.

Response: Thank you for this important hint. We agree with the reviewer that additional data on response times would have been even more informative than data on recognition rates on their own. Due to the implementation as an online study, we were not able to collect reaction times as these would have been highly dependent on the internet connection of the individual participants and thus would probably not reflect their “true” response time. We have specified what is meant by “more difficult” in the introduction (page 7) and have acknowledged possible speed-accuracy trade-offs in the discussion, please see page 23:

Introduction – page 7:

“We expected impaired emotion recognition, i.e. lower recognition rates, for masked faces compared to unmasked faces. Additionally, we were interested in whether wearing a face mask affects the perception of threat and therefore included an exploratory part where we assumed that faces with a mask would be perceived as more threatening than faces that are fully visible.”

Discussion – page 23:

“Moreover, due to the nature of online-assessment, we were not able to collect data on response times in the current study but instead relied on measuring accuracy, i.e. the number of correct responses. Future studies assessing emotion recognition performance should consider collecting response times as well, which would allow for assessment of possible speed-accuracy trade-offs.”

8) Sample: what was the reasoning behind including older adults in the sample? Older adults may have differences in emotional face recognition compared to younger adults.

Response: Thanks for raising this issue. We assessed participants within a wide age range, as we were interested in the effect of face masks on emotion recognition in a sample representative of the general population. To check whether there is an association of participant age with emotion recognition accuracy (and, in parallel, with perception of threat) for masked and unmasked faces, we have now included linear regression analyses. The analyses revealed a significant association of age with emotion recognition accuracy (and perception of threat) for masked faces: Older participants had lower recognition rates for faces with a mask and rated masked faces as less threatening. We did not find an effect of participant age on recognition accuracy (or rating of threat) of unmasked, i.e. fully visible, faces. Please see the respective sections in the results (page 14 and page 16) and discussion sections (page 19):

Results facial emotion recognition – page 14:

“One factor that has been found to influence emotion recognition is the age of participants with older adults showing differences in recognition accuracy compared to younger adults (39,41,42). To check whether there is an effect of participant age on emotion recognition accuracy for masked and unmasked original faces, we performed linear regression analyses. Results from the analysis on the association of participant age with emotion recognition accuracy for masked faces revealed a significant effect of age, F(1, 788) = 30.21, p < .001, R2 = .04, with lower recognition rates for faces with a mask for older participants. In contrast, the analysis on the association of participant age with recognition rates for unmasked original faces did not reveal a significant effect, F(1, 788) = 1.38, p = .241. See S2 Table for emotion-specific results.”

Results perception of threat – page 16:

“To check whether there is an effect of participant age on the perception of threat for masked and unmasked original faces (in parallel to the effect of age on emotion recognition accuracy), we performed linear regression analyses. Results from the analysis on the association of participant age with rating of threat for masked faces revealed a significant effect of age, F(1, 788) = 12.87, p < .001, R2 = .02, with older participants rating faces with a mask as less threatening. In contrast, the analysis on the association of participant age with perception of threat for unmasked original faces did not reveal a significant effect, F(1, 788) = 1.74, p = .188. See S5 Table for emotion-specific results.”

Discussion – page 19:

“Furthermore, we found an association of participant age with emotion recognition accuracy for masked faces: Recognition rates were lower for older participants. We did not find an association of age with recognition accuracy for unmasked faces. Prior research investigating age differences in emotion recognition performance for fully visible faces report inconsistent emotion-specific effects: while emotion recognition has been found to be less accurate for fear (39,41), anger (39,41,42) and sadness (41–43) in older participants (between 65-80 years of age), improved recognition with older age (between 58-70 years) has been observed for happiness (43) and disgust (42). Contrary to previous studies, we did not find a significant effect of age in emotion recognition for faces without a face mask, i.e. fully visible faces. This might be due to previous research investigating age differences by analysing age groups (39,41–43) instead of considering age as a continuous variable, as we did in the present study.

Similar to the findings on age differences in emotion recognition, we observed an association of participant age with perception of threat for masked faces but not for unmasked faces: Older participants rated masked faces as less threatening. This might be due to older participants having experienced face masks as more of a precautionary measure protecting from disease rather than as a health threat.”

9) P 7, 169: Clarify whether participants had an unlimited time to view each image, and whether the response options were presented at the same time as the image.

Response: Thanks. In the experiments, we presented one image as well as the response options and Visual Analogue Scale on one page at a time. There was no time pressure for participants to identify the emotion depicted. We have specified this in the methods section (please see page 10):

“In both experiment 1 and 2, participants were instructed to recognise the respective emotion depicted and to select their answer in a forced-choice answering format where all categories were present (anger, fear, happiness, sadness, disgust and neutral). Each image was presented for an unlimited amount of time and the response options were presented at the same time as the image. There was no time limit for giving a response.”

10) P 9, 250: If I’m reading it correctly, it seems odd that there are no differences in rated threat between emotions, given that both happy and angry expressions are presented. Is this threat measure sensitive?

Response: Thank you for commenting on this point. Please note that in the addressed section on the effect of the presence of a mask vs. no mask on perceived threat (formerly on page 9), we originally only reported the main effect of emotion but not the separate ANOVAs comparing perception of threat between all emotions separately. Further, we only reported results for the difference scores and not for the mean scores separately for the ‘mask’ and ‘original’ conditions.

We performed a sensitivity check for the threat measure by analysing whether there are differences in rated threat between emotions in the ‘original’ condition only. The ANOVA revealed that rating of threat for each emotion was significantly different from rated threat for each other emotion (all p < .05), please also see page 16. We therefore conclude that the threat measure is indeed sensitive and that, as you would expect, e.g. angry faces were rated as significantly more threatening than happy faces.

“In the original condition, angry faces were rated most threatening, followed by fearful, disgusted, sad, neutral and, lastly, happy expressions. In the mask condition, anger was rated most threatening, followed by disgust, fear, sadness, neutral and, lastly, happiness. Rated threat for each emotion was significally different from rated threat for each other emotion, both for masked and unmasked faces (all p < .05).”

11) P 10, 271-275: It’s implied that all expressions are affected by masks, but that’s not what has been found in the results (where happy, fearful and neutral faces were not significantly affected).

Response: We thank the reviewer for making us aware of this inaccurate phrasing. After removing former Study 1 and presenting all data together in one large ANOVA (omitting the analysis of difference scores and instead using mean recognition scores), we actually do find that all expressions are affected by masks. We have rephrased and specified the findings accordingly (please see page 18):

“The findings from experiment 1 suggest that emotion recognition is more difficult for faces wearing a face mask covering mouth and nose than for uncovered faces. In line with our hypothesis, we observed that emotion recognition was less accurate when faces were presented with a mask, for all emotions tested (anger, fear, happiness, sadness, disgust and neutral expressions).”

12) P 12, 360: State whether between or within participant variables were entered into the ANOVA.

Response: Thank you for pointing out this missing information. We have added the information on whether between or within participant variables were entered into the ANOVA accordingly, see page 11:

“For this purpose, we relied on a 6 x 4 repeated-measures analysis of variance (ANOVA) with emotion (anger, fear, happiness, sadness, disgust and neutral) as within-subject factor and type of occlusion (half, bubble, mask and original) as between-subjects factor.”

13) P 17, 522: I don’t think we should assume that German speaking individuals are white. I would be more comfortable if this sentence was removed from the manuscript.

Response: We thank the reviewer for making us aware that the assumption made in the mentioned sentence in the manuscript may be false and we apologise for the inconvenience. We fully agree that this assumption is inappropriate and we have removed the respective sentence and rephrased the paragraph (please see page 23):

“Another limitation of the present research is that, while the stimulus material used was validated for a German-speaking, i.e. possibly mostly Western, population, we did not assess the participants’ ethnicity or cultural background. Previous studies have shown that recognition of basic emotions might actually not be culturally universal, as often assumed (44), but that facial expressions may be perceived and decoded differently dependent on the observer’s culture and may especially differ between the Western and Eastern culture (45). While the vast majority of the study samples in the present research was German-speaking (Study 1: 94%, Study 2: 100%), we were not able to confirm that all participants were Caucasian or raised in a Western cultural background. Future studies should consider assessing ethnicity, as assessment of the individual cultural background could inform about how accepting and familiar the observer is with face masks, as in East Asia, for instance, it has been customary to wear a mask in public for quite some time.”

---

## [Decision Letter · Decision Letter 1]

26 Nov 2021

PONE-D-21-12703R1The impact of face masks on emotion recognition performance and perception of threatPLOS ONE

Dear Dr. Grahlow,

Thank you for submitting your manuscript to PLOS ONE. After careful consideration, we feel that it has merit but does not fully meet PLOS ONE’s publication criteria as it currently stands. Therefore, we invite you to submit a revised version of the manuscript that addresses the points raised during the review process.

Reviewer #2 and I have read the revision of your manuscript and your responses to the reviews. In our opinion, the quality of your work is substantially improved and you were responsive to most of the comments. However, there are some points left which should be adressed in a further revision. Some restructuring of the manuscript is needed (see for details the attached comments of reviewer #2). The abstract should be updated and age should already be mentioned in the introduction section as an important factor influencing facial emotion recognition. Your conclusions need also some rewording. Please check the access to the raw data of your investigation.

We look forward to receiving your revised manuscript.

Kind regards,

Thomas Suslow, Ph.D.

Academic Editor

PLOS ONE

Journal Requirements:

Reviewers' comments:

Reviewer's Responses to Questions

**Comments to the Author**

1. If the authors have adequately addressed your comments raised in a previous round of review and you feel that this manuscript is now acceptable for publication, you may indicate that here to bypass the “Comments to the Author” section, enter your conflict of interest statement in the “Confidential to Editor” section, and submit your "Accept" recommendation.

Reviewer #2: (No Response)

2. Is the manuscript technically sound, and do the data support the conclusions?

Reviewer #2: Yes

3. Has the statistical analysis been performed appropriately and rigorously? 

Reviewer #2: Yes

4. Have the authors made all data underlying the findings in their manuscript fully available?

Reviewer #2: No

5. Is the manuscript presented in an intelligible fashion and written in standard English?

Reviewer #2: Yes

6. Review Comments to the Author

Reviewer #2: The authors have responded to most of my comments very well. I have a few new comments, mostly driven by the changes that have been made to the manuscript.

The structure of the manuscript could use some work now the results are analysed in one ANOVA. I don’t think it makes sense to present them as Experiment 1 and Experiment 2 in the methods, but analyse the experiments together. I would recommend introducing them as different conditions that were between versus within participant and then cluster the results by hypothesis (as they currently are). It can be clear that they were carried out in a serial order, so the methods and results can still capture the intention with which they were run. The abstract should also be updated in this way.

Given that age now seems to be explored in the results, it would be helpful to have some context related to this in the introduction. Line 310: for the regression with age, were responses collapsed across emotions? It could be clearer.

The conclusion seems to be mainly based on evidence that wasn’t provided in the current study (i.e. discussing the possibility of face masks hindering emotion perception in different disorders). Line 580: In many cases it’s not our choice whether to accept the consequences of face masks (they have been mandated in much of the world), therefore this final sentence needs rewording.

In addition, I attempted to access the raw data from the DOI provided, but was unable to, with an error saying to try again later. It may be a problem at my end, but would be helpful if the authors could double check.

7. PLOS authors have the option to publish the peer review history of their article (what does this mean?). If published, this will include your full peer review and any attached files.

Reviewer #2: No

---

## [Author Response · Author response to Decision Letter 1]

3 Jan 2022

We would like to thank the reviewer and the Academic Editor for their constructive and insightful comments that helped improving our manuscript once more. Please find our response in detail below. Any changes within the manuscript are highlighted in yellow.

Academic Editor

 Thank you for submitting your manuscript to PLOS ONE. After careful consideration, we

 feel that it has merit but does not fully meet PLOS ONE’s publication criteria as it currently

 stands. Therefore, we invite you to submit a revised version of the manuscript that

 addresses the points raised during the review process.

 Reviewer #2 and I have read the revision of your manuscript and your responses to the

 reviews. In our opinion, the quality of your work is substantially improved, and you were

 responsive to most of the comments. However, there are some points left which should

 be addressed in a further revision. Some restructuring of the manuscript is needed (see

 for details the attached comments of reviewer #2). The abstract should be updated, and

 age should already be mentioned in the introduction section as an important factor

 influencing facial emotion recognition. Your conclusions need also some rewording. Please

 check the access to the raw data of your investigation.

Response: Thank you very much for the opportunity to revise our manuscript and to resubmit once more. We appreciate the acknowledgement of our work so far and have paid special attention to address the remaining points raised.

Journal Requirements

Please review your reference list to ensure that it is complete and correct. If you have cited papers that have been retracted, please include the rationale for doing so in the manuscript text or remove these references and replace them with relevant current references. Any changes to the reference list should be mentioned in the rebuttal letter that accompanies your revised manuscript. If you need to cite a retracted article, indicate the article’s retracted status in the References list and also include a citation and full reference for the retraction notice.

 Response: Thank you for this comment. We have reviewed our reference list and ensured

 that it is complete and correct to the best of our knowledge. None of the papers cited have

 been retracted.

Reviewer 2

 The authors have responded to most of my comments very well. I have a few new

 comments, mostly driven by the changes that have been made to the manuscript.

 1) The structure of the manuscript could use some work now the results are analysed in

 one ANOVA. I don’t think it makes sense to present them as Experiment 1 and Experiment

 2 in the methods but analyse the experiments together. I would recommend introducing

 them as different conditions that were between versus within participant and then cluster

 the results by hypothesis (as they currently are). It can be clear that they were carried out

 in a serial order, so the methods and results can still capture the intention with which they

 were run. The abstract should also be updated in this way.

 Response: We thank the reviewer for this useful hint and for suggesting a more suitable

 structure for the manuscript. We have restructured the manuscript accordingly and now

 introduce the experimental conditions as two different conditions that were carried out in a

 serial order: mask vs. original (former Experiment 1) and half vs. bubble (former Experiment

 2). Please see the methods section, page 8:

 “The current study comprised two experimental conditions. Condition mask vs. original

 (with stimuli ‘mask’ and ‘original’; please see Fig 1) was conducted by 790 participants (636

 women) between 18 and 89 years (M = 30.85, SD = 12.68). Condition half vs. bubble

 consisted of two parallel versions using the stimuli ‘half’ and ‘bubble’, see Fig 1. Here, 395

 participants (289 women) between 18 and 70 years (M = 29.16, SD = 11.12) saw ‘half’ faces

 and 388 participants (298 women) between 18 and 65 years (M = 28.94, SD = 10.83) were

 presented with ‘bubble’ stimuli.”

 We have also updated the abstract in this regard:

“Facial emotion recognition is crucial for social interaction. However, in times of a global pandemic, where wearing a face mask covering mouth and nose is widely encouraged to prevent the spread of disease, successful emotion recognition may be challenging. In the current study, we investigated whether emotion recognition, assessed by a validated emotion recognition task, is impaired for faces wearing a mask compared to uncovered faces, in a sample of 790 participants between 18 and 89 years (condition mask vs. original). In two more samples of 395 and 388 participants between 18 and 70 years, we assessed emotion recognition performance for faces that are occluded by something other than a mask, i.e., a bubble as well as only showing the upper part of the faces (condition half vs. bubble). Additionally, perception of threat for faces with and without occlusion was assessed. We found impaired emotion recognition for faces wearing a mask compared to faces without mask, for all emotions tested (anger, fear, happiness, sadness, disgust, neutral). Further, we observed that perception of threat was altered for faces wearing a mask. Upon comparison of the different types of occlusion, we found that, for most emotions and especially for disgust, there seems to be an effect that can be ascribed to the face mask specifically, both for emotion recognition performance and perception of threat. Methodological constraints as well as the importance of wearing a mask despite temporarily compromised social interaction are discussed.”

 2) Given that age now seems to be explored in the results, it would be helpful to have

 some context related to this in the introduction. Line 310: for the regression with age,

 were responses collapsed across emotions? It could be clearer.

 Response: We thank the reviewer for pointing out the missing information in the introduction and the results section. We have added information on the influence of age on emotion recognition in the introduction, please see page 5:

 “One factor that might influence emotion recognition is the participant’s age. Research

 investigating age differences report inconsistent emotion-specific effects: while emotion

 recognition seems to be less accurate for fear [27,28], anger [27–29], and sadness [28–30]

 in older participants (between 65-80 years of age), improved recognition with older age

 (between 58-70 years) has been observed for happiness [30] and disgust [29].”

 Further, we have clarified the description of the regression analysis with age in the results

 section. We present results on the association of age with recognition accuracy for masked

 and unmasked faces in general in the results section, while emotion-specific results can be

 found in S2 Table, please see page 14:

 “To check whether there is an effect of participant age on emotion recognition accuracy for

 masked and unmasked original faces in general as well as for each emotion, we performed

 linear regression analyses. Results from the analysis on the association of participant age

 with emotion recognition accuracy for masked faces revealed a significant effect of age, F(1,

788) = 30.21, p < .001, R2 = .04, with lower recognition rates for faces with a mask for older

 participants. In contrast, the analysis on the association of participant age with recognition

 rates for unmasked original faces did not reveal a significant effect, F(1, 788) = 1.38, p =

 .241. See S2 Table for emotion-specific results.”

 3) The conclusion seems to be mainly based on evidence that wasn’t provided in the

 current study (i.e., discussing the possibility of face masks hindering emotion perception

 in different disorders). Line 580: In many cases it’s not our choice whether to accept the

 consequences of face masks (they have been mandated in much of the world), therefore

 this final sentence needs rewording.

 Response: Thank you for raising this point. We followed the suggestions reviewer #1

 provided during the last review process to frame the research topic of emotion recognition

 of faces with and without mask in a broader context and to emphasise possibly detrimental

 effects of face masks on social interaction for specific groups of individuals. Due to

 economic validity we decided to forego assessing mental health in the current study,

 however, we decided – also stimulated by reviewer #1’s comments – to still discuss our

 findings in the context of mental disorders but drastically shortened this part of the

 discussion, please see page 24:

 “To conclude, we were able to show that wearing a face mask affects social interaction in

 terms of impaired emotion recognition accuracy and altered perception of threat. This might

 be especially relevant and possibly detrimental for individuals with certain mental

 disorders who show altered emotion recognition, for instance individuals with autism

 spectrum disorder, major depressive disorder or alcohol use disorder. For these persons,

 face masks may pose an additional obstacle during social interaction which is already

 impeded due to the nature of their mental disorder [48]. Future studies on the effect of

 face masks on emotion recognition and social interaction should therefore additionally

 focus on measures of social competence and assess whether psychopathology moderates

 the effects of face masks on emotion recognition.”

 Thank you for pointing out that our concluding sentence is not up to date anymore

 considering the evolving pandemic situation. We have rephrased accordingly, please see

 page 25:

 “However, despite the limitations posed by mask wearing and the proposed effect on social

 interaction, there are many positive consequences of wearing face masks, such as touching

 one’s face less often [49], which could in turn prevent the spread of infectious diseases.”

 4) In addition, I attempted to access the raw data from the DOI provided, but was unable

 to, with an error saying to try again later. It may be a problem at my end but would be

 helpful if the authors could double check.

 Response: We thank the reviewer for making us aware that our raw data were not

 accessible. We have double checked the access and found that we could access the raw

 data. Please note that it might be necessary to download the provided csv-files instead of

 viewing the data in the repository itself. Please find the relevant DOI necessary to access our

 data in the Open Science Framework here: https://doi.org/10.17605/OSF.IO/JY2ES.

---

## [Editor Report · Decision Letter 2]

7 Jan 2022

The impact of face masks on emotion recognition performance and perception of threat

PONE-D-21-12703R2

Dear Dr. Grahlow,

We’re pleased to inform you that your manuscript has been judged scientifically suitable for publication and will be formally accepted for publication once it meets all outstanding technical requirements.

Kind regards,

Thomas Suslow, Ph.D.

Academic Editor

PLOS ONE
---

## [Editor Report · Acceptance letter]

3 Feb 2022

PONE-D-21-12703R2 

The impact of face masks on emotion recognition performance and perception of threat 

Dear Dr. Grahlow:

I'm pleased to inform you that your manuscript has been deemed suitable for publication in PLOS ONE. Congratulations! Your manuscript is now with our production department. 

Kind regards, 

on behalf of

Professor Thomas Suslow 

Academic Editor

PLOS ONE